# Cardiovascular Disease and the Female Disadvantage

**DOI:** 10.3390/ijerph16071165

**Published:** 2019-04-01

**Authors:** Mark Woodward

**Affiliations:** 1The George Institute for Global Health, University of Oxford, Oxford OX1 2BQ, UK; markw@georgeinstitute.org.au; 2The George Institute for Global Health, University of New South Wales, Sydney, NSW 2050, Australia; 3Department of Epidemiology, Johns Hopkins University, Baltimore, MD 21287, USA

**Keywords:** women, cardiovascular disease, sex differences

## Abstract

Age-standardised rates of cardiovascular disease (CVD) are substantially higher in men than women. This explains why CVD has traditionally been seen as a “man’s problem”. However, CVD is the leading cause of death in women, worldwide, and is one of the most common causes of disability-adjusted life-years lost. In general, this is under-recognised and, in several ways, women are disadvantaged in terms of CVD. Both in primary and secondary prevention, there is evidence that women are undertreated, compared to men. Women often experience heart disease in a different way compared to men, and lack of recognition of this has been shown to have adverse consequences. Female patients of male cardiac physicians have been found to have worse outcomes than their male counterparts, with no such gender differential for female cardiologists. Clinical trials in CVD primarily recruit male patients, yet, it is well recognised that some drugs act differently in women and men. Diabetes and smoking, and perhaps other risk factors, confer a greater proportional excess cardiovascular risk to women than to men, whilst adverse pregnancies and factors concerned with the female reproductive cycle give women added vulnerability to CVD. However, women’s health research is skewed towards mother and child health, an area where, arguably, the greatest public health gains have already been made, and breast cancer. Hence there is a need to redefine what is meant by “women’s health” to encompass the whole lifecycle, with a stronger emphasis on CVD and other non-communicable diseases. Sex-specific analyses of research data should be the norm, whenever feasible.

## 1. Introduction

When most people talk about the societal and medical problems of coronary heart disease (CHD), the typical image they have is of a man with poor lifestyle habits. For sure, such a man is very likely on his way to a heart attack or stroke, but the same would be true for a woman with similar habits. Yet, when they talk about the health issues of middle-aged women, the immediate thought is of breast cancer. Furthermore, experiences in pregnancy are most often the subject of health research and international grant funding in women’s health.

In reality, cardiovascular disease (CVD, mainly CHD and stroke) is the biggest killer of women across the world and in the majority of countries, including those that are rich and all but the poorest (Table 1). CHD is the leading killer amongst women globally and in high-income countries; it only sits behind lower respiratory tract infections, neonatal disorders, diarrhoea and malaria in low-income countries. Stroke ranks close behind CHD, being the second biggest killer of women globally, third in high-income countries and sixth in low-income countries.

Table 2 uses the same format to show the top 10 causes of disability-adjusted life-years lost (DALYs) in women; that is, it considers morbidity as well as mortality. Globally, CHD and stroke are, respectively, the second and third biggest causes of DALYs. Even in low-income countries, stroke figures in the top 10 causes of female DALYs; CHD is outside this column of the table, in 12th place.

For comparison, Table 3 and Table 4 show the equivalent results for men. Besides the obvious differences due to sex-specific illnesses, cirrhosis of the liver and road traffic injuries appear in the top 10 global causes of deaths for men only. Lung cancer increases in relative importance, and Alzeheimer’s disease moves in the opposite direction, when comparing male to female deaths. Regarding global DALYs, headache is in the top 10 for women alone, whilst self-harm appears in the top 10 for men in high-income countries, but not women. However, in general, the sexes show very similar results for both deaths and DALYs, particularly in terms of the importance of CHD and stroke. As is observed for women, CHD and stroke are both in the top 10 for men in all countries/regions considered except for DALYs in low-income countries. Whereas CHD falls outside the top 10 DALYs in low-income countries for women, stroke falls just outside for men (in 11th position).

Although these results depend on how causes are grouped and defined, and are subject to inaccuracies in recording and assumptions made when dealing with incomplete data in some countries, they nevertheless give a good indication of the fundamental importance of CVD to women (as well as men) across the world. Note also that breast cancer causes fewer female deaths than CHD or stroke and only appears in the top 10 DALYs for high-income countries, although, in this group of nations, lung cancer kills more women than does breast cancer.

## 2. The Male Disadvantage

Age-specific rates of death from CHD across the world show approximately log-linear increases with increasing age for both women and men. The female rates are below the male rates at all ages, although the rates get closer in old age [2]. For example, see Figure 1. Incidence (i.e., fatal or non-fatal) CHD rates in men also exceed those in women—for example, Figure 2 shows data from the UK Biobank [3]. Even in subgroups of the population where women “catch up” somewhat with men (discussed in Section 4.3), male rates of myocardial infarction (MI; heart attack) considerably exceed female rates. Similar results are found for stroke, although the age-specific rates tend to be closer for women and men than they are for CHD.

## 3. Lifetime Risk

When lifetime risks of CVD are considered, there is little difference between the sexes. Indeed, because women live longer, on average, the number of CVD events may well be higher in women than men. Indeed, in most years since 1984, the number of deaths due to CVD has been higher in women than in men in the USA. Data from the Rotterdam Study [4] have been used to estimate the remaining lifetime risk of CVD at age 55; this was 67% (95% confidence interval 65 to 70%) for men and 66% (64 to 69%) for women, after controlling for competing risk from non-CVD death. Separating CHD and stroke, the study authors estimated that there would be 102 fewer female deaths but 70 more female stroke deaths per 1000, compared to male deaths. Women are likely to have their first CVD event later than men, typically by 5–10 years, and that first event is more likely to be stroke than CHD, in contrast to men.

Hence, although women do appear to have some natural advantage over men, given the delayed onset of their CVD, the male disadvantage in CVD is somewhat of an illusion because women catch up, in broad terms, over the lifespan. The remainder of this article will present situations where women have a cardiovascular disadvantage, illustrated by examples. In some cases, this may be due to biological differences, but, in others, the cause may well be rooted in the myth of CVD being a “man’s disease”.

## 4. Manifestations of Female Disadvantage

Three separate areas in which women have been found to be at a disadvantage compared to men can be identified (Figure 3). These will be discussed in turn.

### 4.1. Personal Health Care

It is commonly accepted that endogenous oestrogen during the fertile period of life delays the manifestation of CVD in women [5]. However, epidemiological evidence suggests that there is no difference in the rate of increase of CVD with age at the point of menopause [2,6]. For example, in Figure 1, there is no evidence of an increasing CHD death rate around the ages of 45–55 years in women or of the start of convergence of the rates for women and men. Furthermore, clinical trials have found no overall cardiovascular benefit of exogenous oestrogen in postmenopausal women [7]. Therefore, women may well have false optimism regarding their degree of protection from CVD due to oestrogen, and this may lead them to underestimate their cardiovascular risk.

Despite anecdotal observations that women’s magazines and social media include much about healthy lifestyles, many women still may be unsure of issues regarding their CVD risk. Also, women may be more likely to put their family first than are men, which would compromise their own health. Evidence for these assertions comes from a Canadian survey of women [8] which found:<1/2 knew that smoking was a risk factor<1/4 named high blood pressure (BP) or cholesterol as risk factors<1/3 identified four common symptoms of a heart attack in womenOf those at high risk (based on medical history and risk factors), 62% rated their risk as low to moderate65% claimed they had most influence on their family’s health.

### 4.2. Professional Health Care

As with the general public, medical practitioners are prone to the misconception of CVD being a predominantly male problem [9]. They also may have an unconscious sex bias. A survey of American cardiologists, using simulated patients, found that they were less likely to rate the usefulness of angiography as “high” for women than for men. This implicit bias was related to their perception that women were less risk-tolerant than men [10].

Such inherent bias may manifest in the underuse of screening for CVD risk, as was found in a large study of Australian general practice in which the adjusted odds of a woman being screened for CVD risk factors were 12% lower than for a man [11]. In secondary prevention, Zhao et al. (2017) presented results from patients with a history of CHD recruited from routine outpatient cardiology clinics in 11 countries across Europe, Asia and the Middle East [12]. After adjusting for age, they compared the achievement of CVD risk factor and lifestyle targets between the sexes and found that women did worse than men in achieving guideline-based targets for high-density lipoprotein cholesterol, low-density cholesterol, total cholesterol, glucose, physical activity, obesity and cardiac rehabilitation. Men did worse only for blood pressure control and for achieving the target of not smoking (however, to put the latter result in context, female smoking is unusual in Asian and middle-eastern countries). Overall, only 8% of men reached all treatment targets, but the percentage was even lower for women, at 6%; lifestyle targets were more often met but, still, were achieved by relatively few people: 34% of men and 32% of women. Although individual patients have a great deal of responsibility for their own health behaviour, it is reasonable to conclude that there is more that their carers can do to improve risk factor awareness and action, both overall and in relation to sex disparities.

There is also evidence that the treatment given after a CVD event to a woman may be less adequate than that given to a man, which will contribute to risk factor differences in secondary prevention. For example, US guidelines say that patients who survive an MI should be given high-intensity statins (cholesterol-lowering drugs). Yet, Peters et al. (2018), using data from two large health insurance systems, found that, amongst all patients given statins, American women have been less likely than American men to receive high-intensity statins since 2007 [13]. In 2014/5, 9% (95% confidence interval 8 to 10%) fewer women than men filled a prescription for a high-intensity statin within 30 days, in this study population.

In the acute setting, physicians may miss heart attacks in women because they often experience attacks in a different way compared to men, classic medical text-books having been written according to the male model of the disease. Specifically, women seem to be more prone than men to experience shortness of breath, nausea, vomiting, back, shoulder or jaw pain, and anxiety as symptoms of MI. They are also less likely to experience physical exercise as the trigger to their MI, instead being more likely to experience emotional distress. A UK study found that the odds of a woman having an incorrect initial diagnosis on admission to hospital were 37% higher for women than men, amongst all patients who had a ST-elevation myocardial infarction (STEMI), and 29% higher for patients who had a non-ST-elevated MI (NSTEMI) [14]. Since those with an incorrect initial diagnosis were more likely to die within 12 months than were those with a correct initial diagnosis (time to death was 10% shorter for misdiagnosed STEMI cases and 14% shorter for misdiagnosed NSTEMI, after adjustment), this suggests that misdiagnosis is an extremely important problem, more so in women. Figure 4 shows data from another study, involving 582,000 admissions to emergency departments in Florida [15]. Regardless of the sex of the cardiac physician attending, after allowing for several confounding factors, survival was worse for women than for men. A striking feature of these data is that, although female physicians appear to have similar results for female and male patients, amongst patients treated by men, female patients survive their treatment less often than do male patients. Although the differences in probabilities in Figure 4 are small, this issue, if widespread, is of immense importance in terms of female lives that might be saved and is also compounded by the relative lack of female cardiologists in many settings. To address such issues, women’s cardiology clinics have recently been founded [9] but are yet to have a wide distribution.

For those who survive a MI and leave hospital, women tend to have worse survival and be more likely to have a recurrent event. Figure 5 shows an example, although, in this case, sex differences disappeared after adjusting for age [16]. This is explained by women being generally older when they suffer MI, which shows the importance of controlling for age in such sex comparisons. All the same, given the female advantage in age-specific CHD rates before a CHD event, similar age-adjusted survival after an MI suggests that women have lost some of their natural advantage, which might be explained by women receiving worse secondary prevention care, as discussed above.

The Food and Drug Administration of the USA has a history of more than 20 years of promoting reports of clinical trial results by sex, because experience has shown that some drugs differ in efficacy or safety between the sexes [17,18,19], as acknowledged by the American Heart Association in 2016 [20]. However, such policies are not universal, and there is no sign that cardiovascular trials are yet recruiting as many women as men, even in the USA [19,21,22].

### 4.3. Risk Factors for both Sexes

Sex differences in the prevalence of classical modifiable CVD risk factors—high levels of blood pressure and cholesterol, smoking, diabetes and overweight/obesity—can be expected to have a substantial effect on sex differences in CVD (as well as other clinical) outcomes. These prevalences differ greatly across the world, and it is impossible to draw generalisations. Instead, one contemporary example will be mentioned. This is a recent analysis of national surveys amongst adults in the USA between 2001 and 2016 [23] which found that trends in reductions in systolic blood pressure and smoking and increasing prevalence of diabetes mellitus were similar between women and men. However, reductions in total cholesterol were greater in men, and increases in body mass index were greater in women. As in a global study of obesity [24], women were more likely than men to be obese.

In a systematic series of large-scale meta-analyses of various sets of cohort studies from general populations, several commonly recognised cardiovascular risk factors (smoking [25], diabetes [26], atrial fibrillation [27] and low socioeconomic status [28]) were shown to have a stronger relative effect on CHD in women than in men, whilst only total cholesterol had a stronger effect on CHD in men than women [29]. There was no evidence of a sex difference associated with higher blood pressure [30] or body mass index [31] (Figure 6). Therefore, although all these factors increase the risk of CHD in both sexes, some are associated with additional excess risk in one sex compared to the other.

Parallel meta-analyses, in an overlapping but distinct set of cohort studies, found somewhat similar results for stroke (Figure 7). Again, there was a clear female disadvantage for atrial fibrillation [27] and diabetes [32], but the excess female relative risk for smoking was small (6%) and marginally non-significant [33]. There was insufficient evidence to reliably estimate the effects of social deprivation on stroke [28], as shown by the corresponding wide confidence interval in Figure 7. Both total cholesterol [29] and blood pressure [30] showed no evidence of a sex difference. These analyses of stroke are, however, limited in that subtypes of stroke were not always distinguishable, and thus Figure 7 shows the results for all types of stroke together. Also, this systematic series of meta-analyses has yet to investigate sex differences in the effects of body mass index on stroke. 

The greatest weight of evidence of a female disadvantage is for diabetes [34]. According to the above meta-analyses, having this risk factor more than doubles the risk of CHD but confers an additional 44% risk to women compared to men (see Figure 6). Results for stroke are similar but somewhat less extreme: the excess female risk is 27% (see Figure 7). This could be caused by relatively worse care to prevent the sequelae of diabetes received by women compared to men [35] and by women tending to have greater adiposity when diagnosed as positive for diabetes [36,37].

A general limitation with these meta-analyses is that most of the results that were pooled came from published studies which employed various methods and covariate adjustments. Although all adjusted for age, age-specific sex differences were not routinely reported and could not be reliably summarised. Another limitation is that it was only possible to pool relative risks. Avoiding these issues, whilst still using a large set of data, Millett et al. (2018) analysed sex differences in risk factor associations with MI in the UK Biobank (*n* = 471,988) [3]. The female disadvantage for diabetes and smoking, on the relative scale, seen in Figure 6, was confirmed, but there was also evidence of such a disadvantage from high blood pressure. Results for atrial fibrillation and low (versus high) social status went in the same direction (to women’s relative disadvantage) as in Figure 6 but were not statistically significant. Sex differences, where they occurred, in relative risks were retained with ageing. On the absolute scale, men had higher risk differences than women for hypertension, smoking, overweight/obesity but not diabetes.

### 4.4. Female Risk Factors

Factors concerned with reproduction of the species are clearly specific to women: several have been shown to be associated with future CVD. Peters et al. (2018) found each of the following to increase the risk of CVD in the UK Biobank study population [38]:Early menarcheEarly menopauseHistory of hysterectomy

Amongst those who gave birth:Early age at first birthHistory of miscarriageHistory of stillbirth.

In this analysis, and another using the similar China Kadoorie Biobank [39], there was also evidence of an increasing risk of both CHD and stroke with an increasing number of children, at least in those women who had any children. Although this phenomenon had formerly been thought to be caused by female biological factors, in both biobanks the association between CVD and number of children was found to be very similar in men, leading to a conclusion that the causal factors were most likely social in origin. This shows the great value of including male controls, whenever possible, when researching women’s health. In particular, it illustrates a limitation in drawing inferences from the Canadian survey of women’s cardiovascular knowledge discussed in Section 4.2.

Adverse pregnancies, involving gestational diabetes (GD) or pre-eclampsia (PE), are also risk factors for future CVD in the mother. For example, in the second Nurses Heath Study, after adjusting for age, pre-pregnancy body mass index and other covariates, GD was associated with subsequent CVD, with a hazard ratio (95% confidence interval) for GD versus no GD of 1.43 (1.12 to 1.81) [40]. Also, in a meta-analysis [41], relative risks (95% confidence intervals) for PE versus no PE were: 2.33 (1.95 to 2.78) for CHD; 2.03 (1.54 to 2.67) for stroke; and 2.29 (1.73 to 3.04) for cardiovascular mortality.

A drawback with most studies of pregnancies is that the data used typically derive from maternal registries which lack data on classic CVD risk factors prior to, and after, the pregnancy. Thus, for example, PE could simply reflect a woman’s own natural high blood pressure. This not only requires a caveat to statements about the additional (or “independent”) nature of PE and GD over and above classical CVD risk factors, but also undermines the ability to reliably include PE and GD in a CVD risk score for general use in women currently free of CVD. This presents a fundamental limitation when considering the true CVD risk of women who have a history of adverse health in pregnancy and could thus mean that many fail to receive life-saving treatments.

## 5. Discussion

Whilst it is undoubtedly true that men suffer CVD at an earlier age and at a higher age-specific rate than women, the lifetime risks for both sexes appear to be similar. Also, there are many areas where women have been shown to suffer a cardiovascular disadvantage, as this article illustrates. This could be because of biological, medical or behavioural differences, and a key challenge of sex differences research is to identify such causal processes. All the same, not all sex differences are bad; indeed, sex differences in general are to be celebrated. What is required is research to remove sex inequities in health. This must start with identifying where the problems lie and quantifying the inequities. Unfortunately, much CVD research still under-represents women, which is clearly an obstacle to such aims.

A limitation to this article is that sex and gender have not been distinguished. Broadly, “sex” is generally taken to refer to biological differences, and “gender” to social or cultural differences between women and men. Without prejudice, “sex” alone has been used here, both for simplicity and because most of the material quoted has not made the distinction, generally using only the noun “sex”.

## 6. Conclusions

CVD is not a “man’s disease”. Indeed, it is arguably the most important disease amongst women globally. Despite this, its importance in women is insufficiently recognized. Addressing this gap requires a fundamental change in the way we view women’s health. Four key recommendations are:A broader women’s health agenda is needed, integrating sexual and reproductive health with CVD and other non-communicable diseases—taking a lifecourse approach to women’s healthSufficient numbers of women should be included in studies of CVD (and more generally)A sex-disaggregated approach should be taken to collecting, analysing and reporting CVD (and more generally); given that roughly 50% of the population is female, it seems reasonable that responsible official bodies and journals should mandate that all research (except in extenuating circumstances) should report results by sexWhere appropriate, male controls should be included in studies of women’s health.

## Figures and Tables

**Figure 1 ijerph-16-01165-f001:**
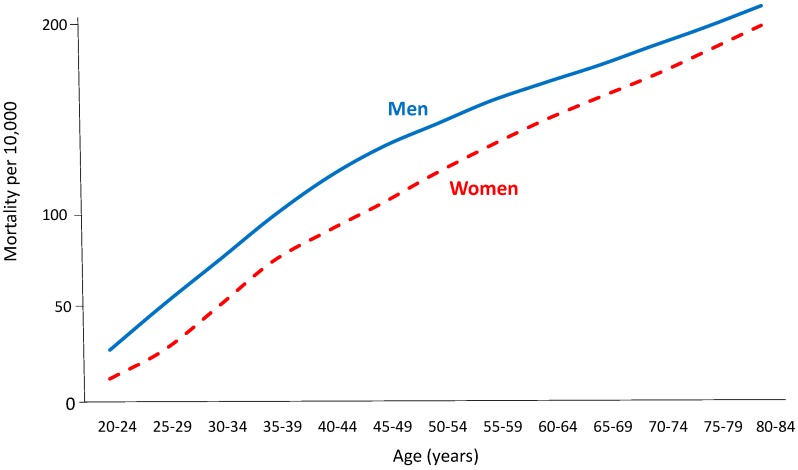
Mortality rates for coronary heart disease in the USA in 2000, by sex; Global Burden of Disease study data.

**Figure 2 ijerph-16-01165-f002:**
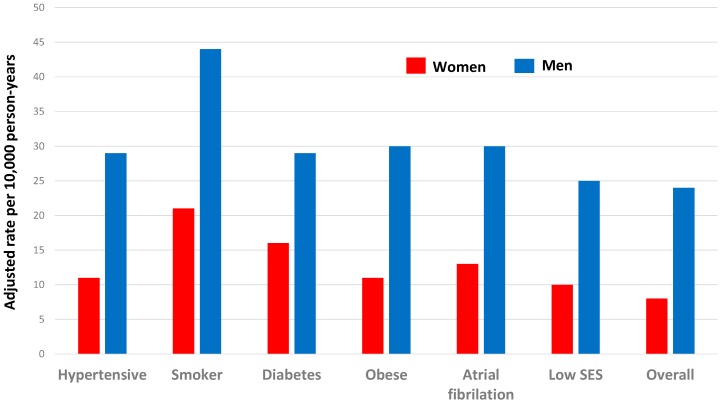
Incidence rates for myocardial infarction amongst people with specific risk factors and overall, in the UK Biobank, by sex, adjusted for age and other CVD risk factors. The study population was 471,988 people without a history of CVD followed up for 7 years [3].

**Figure 3 ijerph-16-01165-f003:**
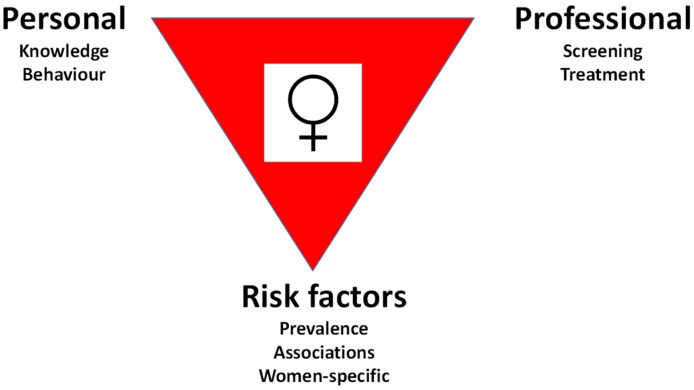
Types of female disadvantage in cardiovascular disease.

**Figure 4 ijerph-16-01165-f004:**
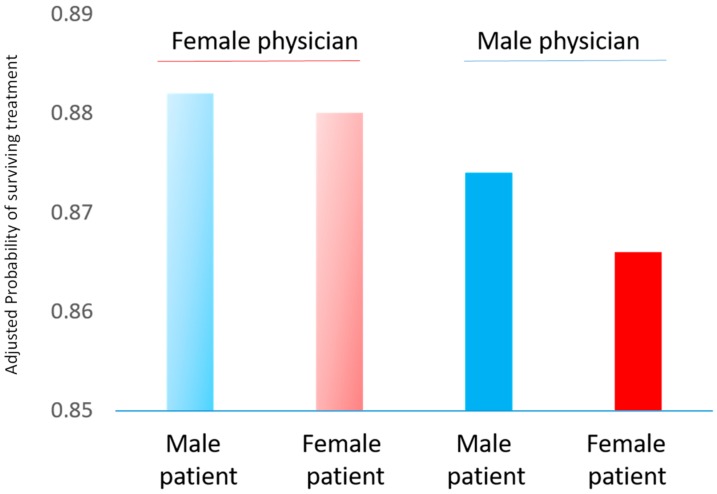
Probability of survival from cardiac treatment in Florida emergency departments, 1991–2010, showing results separately according to the sex of the physician attending and the sex of the patient. This figure was drawn using approximate results derived from the source paper [15].

**Figure 5 ijerph-16-01165-f005:**
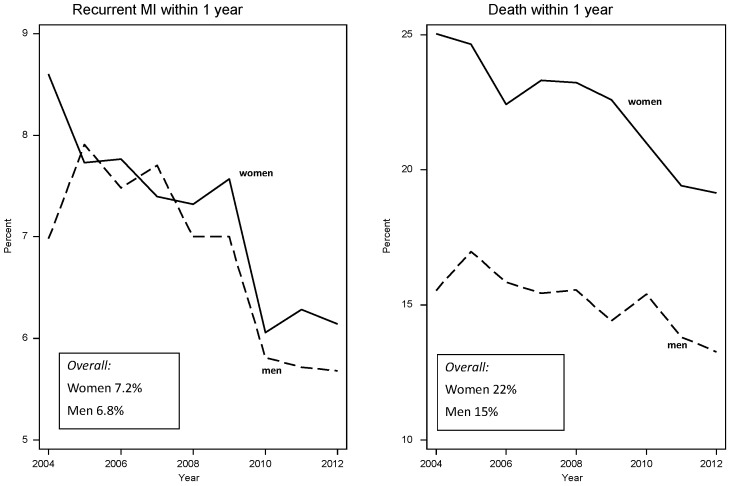
Percentage of myocardial infarction (MI) patients experiencing a recurrent event and percentage of patients dying within one year after discharge from New South Wales hospitals, 2004–2014, by sex (*n* = 89,529).

**Figure 6 ijerph-16-01165-f006:**
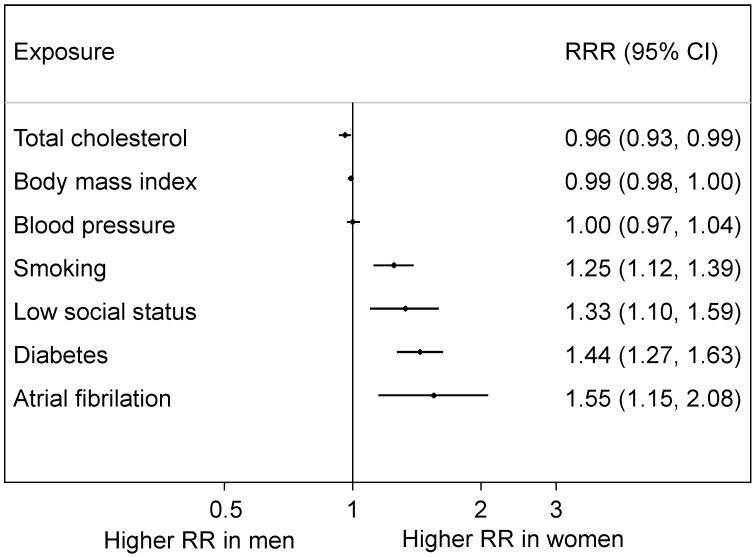
Women-to-men ratios of relative risks (RRRs) for coronary heart disease: pooled results from meta-analyses of numerous individual studies. Total cholesterol, body mass index and blood pressure were all measured continuously (i.e., the relative risks (RRs) are per unit increase); smoking signifies current versus not current; low social status is compared to high social status; diabetes and atrial fibrillation compare yes to no.

**Figure 7 ijerph-16-01165-f007:**
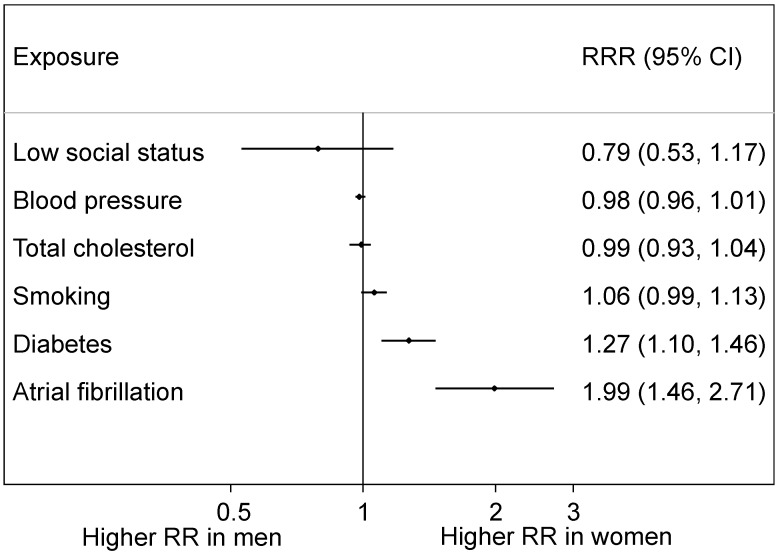
Women-to-men RRRs for stroke: pooled results from meta-analyses. Conventions as in Figure 6.

**Table 1 ijerph-16-01165-t001:** The 10 most common causes of death in women, globally and in selected countries and regions, 2017.

Rank	Global	United States of America	High-Income Countries	India	Low-Income Countries
1	**CHD**	**CHD**	**CHD**	**CHD**	Respir Infect
2	**Stroke**	Alzheimer’s	Alzheimer’s	COPD	Neonatal disorders
3	Alzheimer’s	**Stroke**	**Stroke**	Diarrhoea	Diarrhoea
4	COPD	COPD	Lung cancer	**Stroke**	Malaria
5	Respir Infect	Lung cancer	COPD	Respir Infect	**CHD**
6	Diarrhoea	Breast cancer	Respir Infect	Neonatal disorders	**Stroke**
7	Neonatal disorders	Respir Infect	Breast cancer	TB	HIV/AIDS
8	Diabetes	CKD	Colorectal cancer	Asthma	TB
9	Breast cancer	Colorectal cancer	CKD	Diabetes	Congenital defects
10	Lung cancer	Diabetes	Diabetes	Falls	COPD

Data from The Global Burden of Disease study [1]. Income levels according to the World Bank; CHD = coronary heart disease; TB = tuberculosis; COPD = chronic obstructive pulmonary disease; Respir Infect = lower respiratory tract infection; CKD = Chronic kidney disease. Cardiovascular events are shown in bold.

**Table 2 ijerph-16-01165-t002:** The 10 most common causes of disability-adjusted life-years lost in women, globally and in selected countries and regions, 2017.

Rank	Global	United States of America	High-Income Countries	India	Low-Income Countries
1	Neonatal disorders	**CHD**	Low back pain	Neonatal disorders	Neonatal disorders
2	**CHD**	Low back pain	**CHD**	**CHD**	Respir Infect
3	**Stroke**	COPD	**Stroke**	Diarrhoea	Malaria
4	Respir Infect	Drug use	Alzheimer’s	Respir Infect	Diarrhoea
5	Diarrhoea	**Stroke**	Headaches	COPD	HIV/AIDS
6	COPD	Headaches	COPD	Dietary iron deficiency	Congenital defects
7	Low back pain	Diabetes	Diabetes	**Stroke**	TB
8	Headaches	Lung cancer	Depressive disorders	Headaches	Maternal disorders
9	Diabetes	Depressive disorders	Breast cancer	TB	Malnutrition
10	Congenital defects	Alzheimer’s	Lung cancer	Congenital defects	**Stroke**

For conventions, see Table 1.

**Table 3 ijerph-16-01165-t003:** The 10 most common causes of death in men, globally and in selected countries and regions, 2017.

Rank	Global	United States of America	High-Income Countries	India	Low–Income Countries
1	**CHD**	**CHD**	**CHD**	**CHD**	Neonatal disorders
2	**Stroke**	Lung	Lung cancer	COPD	Respir infect
3	COPD	Alzheimer’s	**Stroke**	**Stroke**	Diarrhoea
4	Respir infect	COPD	Alzheimer’s	Diarrhoea	**CHD**
5	Lung cancer	**Stroke**	COPD	TB	Malaria
6	Neonatal disorders	Respir infect	Respir infect	Respir infect	TB
7	RTI	CKD	Colorectal cancer	Neonatal disorders	**Stroke**
8	Cirrhosis	Drug use	Prostate cancer	RTI	HIV/AIDS
9	Alzheimer’s	Colorectal cancer	Cirrhosis	Cirrhosis	RTI
10	TB	Cirrhosis	Self-harm	CKD	Congenital defects

For most conventions, see Table 1. RTI = road traffic injuries.

**Table 4 ijerph-16-01165-t004:** The 10 most common causes of disability-adjusted life-years lost in men, globally and in selected countries and regions, 2017.

Rank	Global	United States of America	High-Income Countries	India	Low–Income Countries
1	**CHD**	**CHD**	**CHD**	Neonatal disorders	Neonatal disorders
2	Neonatal disorders	Drug use	Low back pain	**CHD**	Respir infect
3	**Stroke**	COPD	Lung cancer	COPD	Diarrhoea
4	Respir infect	Diabetes	**Stroke**	Respir infect	Malaria
5	RTI	Lung cancer	Diabetes	Diarrhoea	HIV/AIDS
6	COPD	Low back pain	COPD	TB	Congenital defects
7	Diarrhoea	RTI	Drug use	**Stroke**	TB
8	Diabetes	**Stroke**	Self-harm	RTI	RTI
9	Congenital defects	Self-harm	RTI	Diabetes	**CHD**
10	Low back pain	Cirrhosis	Falls	Self-harm	Meningitis

For conventions, see Table 1 and Table 3.

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
