# Peer review of "Cardiovascular Disease and the Female Disadvantage"

_ijerph, 2019, doi:10.3390/ijerph16071165_

Round 1

Reviewer 1 Report

An excellent review outlining the many aspects of CVD where differences have been shown between males and females. Female disadvantage in CVD is clearly an important worldwide, this review contributes towards highlighting this disadvantage and outlining reasons why this occurs. A key conclusion of the paper indicates that sex specific analysis of research data should be the norm.

Many important topics are covered within the review, I read section 4 ‘manifestations of female disadvantage’ with great interest. It is certainly an area of concern that many women are unsure of their CVD risk.

An insightful review of the literature in this area is provided, highlighting key references in the area. Section 4.3 highlights a number of meta-analyses that have been undertaken on the risk factors associated with CVD, and the respective sex differences.

A minor point regarding the graphics, if possible, consider formatting figure 1 and 2 so that their grey scale differs (or line type), for printing in black and white.

Author Response

1. I agree that grey scales are easier to read in black and white but they are less attractive in colour, and I would expect most readers to view this on a computer screen. Therefore I have not used grey scale, but instead have changed both figures to make it more obvious which data are for women and which for men.

Reviewer 2 Report

This review article discusses about the female disadvantage in treatment of cardiovascular disease regarding personal knowledge/behavior, professional screening/treatment and multiple risk factors including gender-specificity. It is informative and clear and overall quality is good. I have several questions for the authors to address and improve for future publication.

In the abstract, the statement "clinical trials in CVD primarily recruit male patients" may be problematic. As far as I know, for example, the Precision Trial conducted by Cleveland Clinic recruit even more number of female patients compared to males and have no gender bias. This might be true for many other trials and thus references need to be provided to support this statement.

Regarding Table 1, have similar analysis been done for men and what might be the conclusion comparing to women results? This information could be put in supplemental information to allow the readers to have more complete picture of causes of death and DALY in both genders.

In Figure 1, the data are shown for year 2000 which might be a little out of date. Is it possible to update it using data for 2017 as what has been shown for Table 1 and 2?

On Page 5, the statement "many women still may be unsure of issues regarding their CVD risk". More references about the situation in other countries and area around the world need to be provided to support this point.

In Figure 5, if adjusting for age, will any difference be observed between the results with and without age adjustment?

In Figure 6, please list what study data base are included and basic patient enrollment criteria for each study.

In Figure 7, are the same pooled meta-analyses database used as the one used in Figure 6? If not, please list the studies you included.

Author Response

As requested, I have added 2 new references to prove my assertion. The PRECISION trial actually falls outside the remit of my paper as it was not applied to patients with CVD, nor did it investigate cardiovascular drugs. 

I have added two new tables, with male results, as suggested.

Unfortunately the data source does not provide recent data by 5-year age groups, but a reader can see fairly recent data in graphical form in the paper I have cited as reference 2. For my purpose, of illustrating age-specific trends, the year 2000 is adequate.

Unfortunately I don't have any other examples, which is why I was careful to use the word "may" in this regard.

Yes, I have now changed the wording to make this clear, thank-you.

There are scores of studies involved, and each meta-analysis had different study bases, as I have now explained in my revised MS. Thus it is unreasonable to list them all here, when the source materials are all cited. Patient enrollment is now described (they were all general population studies).

I have now clarified that CHD and stroke used different study bases. As in #6, there are too many studies to list.

Reviewer 3 Report

The manuscript ‘Cardiovascular disease and female disadvantage’ presents a topic that has been discussed by cardiologists and public health professionals for some time now. So this is not a new topic, but it is well presented. The manuscript was written in the form of a review in which the author describes the epidemiological factors of cardiovascular diseases in women and compares them to a group of men. These comparisons are unfavorable for women, especially at a late age, when coronary mortality rates for women and men are nearly equal. The author distinguishes and discusses three separate areas in which women are disadvantaged: personal health care, professional health care and risk factors.

Introduction. The introduction is supposed to attract our attention to the issue, but the initial verses are more suited to a popular science article. This is better avoided in scientific work.

An interesting issue in relation to women is incorrect medical diagnosis. For example, more women than men are misdiagnosed in the initial diagnosis of heart attack. In fact, women have a higher chance of misdiagnosis in both STEMI and NSTEMI. Pay attention to the correctness of the data provided. Check line 156 on page 6, if the data given are correct in relation to publication [14] and check the odds ratio. The odds are exactly 37% higher for women, but not lower (Fig. 1 in [14]) and this group is treated as a reference (the reference value is 1) for men who have lower odds than women.

Conclusions. The conclusions adequately summarize the issues described. The recommendations come from the observations made by the author.

The issues raised in the manuscript are presented in an appropriate and communicative way. Relevant references are given. Due to the subject matter and form of presenting the problem, it can be published.

Author Response

I have revised the introductory paragraph so as to be less colloquial.

Thank-you for pointing out this silly mistake, which I have rectified.

Round 2

Reviewer 2 Report

The author has fully addressed my concerned and I suggest acceptance.